# A Placebo-Controlled Trial of Cannabinoid Treatment for Disruptive Behavior in Children and Adolescents with Autism Spectrum Disorder: Effects on Sleep Parameters as Measured by the CSHQ

**DOI:** 10.3390/biomedicines10071685

**Published:** 2022-07-13

**Authors:** Aviad Schnapp, Moria Harel, Dalit Cayam-Rand, Hanoch Cassuto, Lola Polyansky, Adi Aran

**Affiliations:** 1Department of Pediatrics, Hadassah Medical Center, Jerusalem 91120, Israel; aviad.schnapp@gmail.com; 2Neuropediatric Unit, Shaare Zedek Medical Center, Jerusalem 9103102, Israel; moriaharel@gmail.com (M.H.); dalitc@szmc.org.il (D.C.-R.); lolapol@szmc.org.il (L.P.); 3Child Development Centers, Leumit Health Services, Jerusalem 9439221, Israel; han.cass@gmail.com; 4Faculty of Medicine, Hebrew University of Jerusalem, Jerusalem 9112102, Israel

**Keywords:** autism spectrum disorder, cannabinoids, cannabidiol, tetrahydrocannabinol, clinical trials, randomized controlled, sleep, child psychiatry, developmental disorders

## Abstract

Autism spectrum disorder (ASD) is often associated with debilitating sleep disturbances. While anecdotal evidence suggests the positive effect of cannabinoids, randomized studies are lacking. Here, we report the effects of cannabinoid treatment on the sleep of 150 children and adolescents with ASD, as part of a double-blind, placebo-controlled study that assessed the impact of cannabinoid treatment on behavior (NCT02956226). Participants were randomly assigned to one of the following three treatments: (1) whole-plant cannabis extract, containing cannabidiol (CBD) and Δ9-Tetrahydrocannabinol (THC) in a 20:1 ratio, (2) purified CBD and THC extract in the same ratio, and (3) an oral placebo. After 12 weeks of treatment (Period 1) and a 4-week washout period, participants crossed over to a predetermined, second 12-week treatment (Period 2). Sleep disturbances were assessed using the Children’s Sleep-Habit Questionnaire (CSHQ). We found that the CBD-rich cannabinoid treatment was not superior to the placebo treatment in all aspects of sleep measured by the CSHQ, including bedtime resistance, sleep-onset delay, and sleep duration. Notably, regardless of the treatment (cannabinoids or placebo), improvements in the CSHQ total score were associated with improvements in the autistic core symptoms, as indicated by the Social Responsiveness Scale total scores (Period 1: r = 0.266, *p* = 0.008; Period 2: r = 0.309, *p* = 0.004). While this study failed to demonstrate that sleep improvements were higher with cannabinoids than they were with the placebo treatment, further studies are required.

## 1. Introduction

Autism spectrum disorder (ASD) is a heterogeneous neurodevelopmental disorder that is characterized by persistent deficits in social interaction and communication, restricted interests, and repetitive behaviors [1]. Phenotypes among individuals with ASD are highly diverse in terms of cognition, language abilities, irritability, sensory perception, anxiety, motor skills, executive functions, epilepsy, gastrointestinal problems, and more [1,2]. Up to 80% of children with ASD also have sleep disorders, including prolonged sleep onset latency, extended night awakenings, and early morning awakenings [3,4].

The etiology of sleep disorders in ASD is presumed to include multiple neuropsychological factors [5], among which alterations in the circadian sleep–wake cycle are the most well established [6,7]. Accordingly, the most common pharmacological treatment for sleep disorders in individuals with ASD is exogenous melatonin, given as an add on to behavioral interventions, and parental education [8].

One of the main regulators of the sleep–wake cycle is the endocannabinoid system [9]. The primary components of this cell-signaling system are the cannabinoid receptors and their endogenous ligands (endocannabinoids) [10]. The cannabis plant contains unique compounds (phytocannabinoids) that can interact with the endocannabinoid system either directly, using Δ9-tetrahydrocannabinol (THC), or indirectly, using cannabidiol (CBD). THC is the major psychoactive component of the cannabis plant. It activates the type 1 cannabinoid receptor (CB_1_R) in the brain and might lead to anxiety and psychosis [11]. CBD is the major non-psychoactive phytocannabinoid. While it is an allosteric modulator of the CB_1_R, which may decrease the effects of CB_1_R agonists, it concomitantly increases the levels of the endocannabinoids that activate the CB_1_R (Figure 1) [11]. As opposed to THC, CBD has a relatively high toxicity threshold and it also appears to have anxiolytic, antipsychotic, antiepileptic, and neuroprotective properties that may be mediated through receptors, such as serotonin 5-HT_1A_, TRPV1, GPR55, GABA_A_, and PPARγ, and through the inhibition of adenosine reuptake (Figure 1) [12,13,14,15,16].

Alterations in the endocannabinoid system have been found in several animal models of ASD [17,18]. Recent human studies have demonstrated lower circulating endocannabinoid levels in children with ASD [19,20,21] and evidence of successful CBD-rich cannabinoid treatment for the core symptoms and comorbidities in children with ASD is accumulating [22,23,24,25]. However, the effect of phytocannabinoids on the sleep of ASD children is still unclear.

In general, cannabinoid therapy with various THC to CBD ratios is being increasingly used to alleviate sleep disorders, regardless of the cause. Among patients with chronic pain, treatment with medical cannabis seems to result in a small improvement in sleep quality [26]. Preliminary evidence of successful treatment is also available for sleep apnea, and post-traumatic stress disorder-related nightmares [27]. However, currently there is insufficient evidence to support this line of treatment for any individual sleep disorder [27,28,29].

In the current study, we aimed to evaluate the impact of a CBD-rich cannabinoid treatment on sleep, as part of a placebo-controlled trial, which assessed the effects of cannabinoids on the behavior of children and adolescents with ASD. We used two CBD-rich preparations. The first contained only purified CBD and purified THC isolates (pure cannabinoids) and the second contained a full-spectrum (whole-plant) extract, which, in addition to the same amounts of CBD and THC, also contained minor cannabinoids, terpenes, and flavonoids that might enhance the efficacy and tolerability.

We found that an improvement in sleep (after receiving either cannabinoids or the placebo) was associated with an improvement in the autistic core symptoms and disruptive behavior. However, CBD-rich cannabinoid treatment did not improve sleep disturbances more than the placebo treatment.

The main cannabinoid receptor in neurons is cannabinoid receptor type 1 (CB_1_R). The primary neuronal effect of the CB_1_R is a decrease in the synaptic transmission during increased synaptic activity, which can also promote sleep and reduce seizures and excitotoxicity. CBD is a negative allosteric modulator of the CB_1_R. However, CBD can activate the endocannabinoid system through the CB_1_R by inhibiting the endocannabinoid membrane transporter (EMT) and the degradation of anandamide (AEA) through fatty acid amide hydrolase (FAAH). This, in turn, increases the levels of the endocannabinoids AEA (main agonist of CB_1_R) and 2-Arachidonoylglycerol (2-AG). Other neuronal effects of CBD are mediated through agonism at the 5-HT_1A_ serotonin receptors and at the TRPV1 channel, reducing anxiety and pain, and through agonism of the nuclear PPARγ receptors, increasing the expression of the cytoprotective enzymes. CBD also has direct antioxidative effects.

ENT—equilibrative nucleotide transporter; 5-HT_1A_—5-hydroxytriptamine 1A receptor; GPR55—G protein coupled receptor 55; PPARγ—peroxisome proliferator-activated receptor gamma; ROS—reactive oxygen species; TRPV1—transient receptor potential vanilloid 1.

## 2. Materials and Methods

### 2.1. Study Design

NCT02956226 was a proof-of-concept, randomized, double-blind, placebo-controlled trial, and methods were previously described [23]. The primary objective of this trial was to assess the impact of cannabinoid treatment on ASD-associated disruptive behavior. We previously reported the effects of the cannabinoid treatment on disruptive behavior and the ASD core symptoms, as well as adverse effects of the treatment [23]. We report here the effect of the cannabinoid treatment on sleep parameters.

### 2.2. Standard Protocol Approvals and Patient Consent

The study was conducted in a single referral center for ASD diagnosis and treatment: Shaare Zedek Medical Center, Jerusalem, Israel. It was approved by the Institutional Review Board at Shaare Zedek Medical Center and the Israeli Ministry of Health prior to participant enrollment. Participants’ parents provided written informed consent and written consent was also obtained from participants, when appropriate.

### 2.3. Study Population

Eligible participants were children and adolescents between 5 and 21 years old, with an ASD diagnosis, as per the Diagnostic and Statistical Manual of Mental Disorders, fifth edition (DSM-5), criteria, and as confirmed by the Autism Diagnostic Observation Schedule (ADOS-2), and moderate or greater behavioral problems rating (rating ≥ 4) on the Clinical Global Impression’s (CGI) severity scale. The full list of inclusion and exclusion criteria appears in Appendix A.

### 2.4. Treatment Scheme

Participants were randomly allocated for treatment with two out of three oral preparations, each given in a distinct 12-week treatment period. Treatment options were as follows: (1) BOL-DP-O-01-W (BOL Pharma, Revadim Israel), a whole-plant (full spectrum) cannabis extract, containing CBD and THC at a 20:1 ratio; (2) BOL-DP-O-01 (BOL Pharma, Revadim, Israel), purified CBD and THC at the same ratio; and (3) placebo (BOL Pharma, Revadim, Israel). In each treatment period, starting dose was 1 mg/kg/d CBD (and 0.05 mg/kg/d THC) or an equivalent placebo. The dose was increased by 1 mg/kg/d CBD (and 0.05 mg/kg/d THC) every other day, up to 10 mg/kg body weight per day CBD (and 0.5 mg/kg/d THC), for children weighing 20–40 kg or 7.5 mg/kg/d CBD (and 0.375 mg/kg/d THC) for weight >40 kg (maximum 420 mg CBD and 21 mg THC per day), divided into 3 daily doses. Treatments were given orally (sublingual whenever possible), as an add-on to any ongoing stable medication (Table 1). At the end of the first treatment period, the study treatment was gradually decreased over 2 weeks, followed by 2 weeks of no study treatment to enable full elimination of the cannabinoids given in the first period [30].

The CBD:THC ratio and daily dose were chosen based on our clinical experience and previous open-label studies on the effect of medical cannabis on ASD core symptoms and comorbidities, including sleep problems [22,24,25]. Further details regarding the cannabinoids’ preparations and randomization process appear in the Appendix A.

### 2.5. Baseline Evaluations

Baseline assessments at study onset (day 1) included the following: ADOS-2 [31], a systematic and standardized assessment of communication, social interaction, play, and imaginary use of materials, which was administered by a developmental psychologist (MH), with research reliability; Vineland Adaptive Behavior Scales (VABS) [32], a caregiver interview assessing communication, socialization, and daily living skills, which was administered by the same psychologist; and Childhood Autism Rating Scale, second edition (CARS2-ST) [33]—A quantitative measure of direct behavior observation–which was administered by a trained pediatric neurologist (AA).

### 2.6. Outcomes

Children’s Sleep Habits Questionnaire (CSHQ) [34]. This parent-rated questionnaire has been used and validated in multiple studies of ASD [35,36,37,38,39]. It comprises 33 scored questions, and additional items intended to provide other relevant information on sleep behavior. Each scored question is rated on a 3-point scale, as occurring ‘usually’ (i.e., 5–7 times within the past week), ‘sometimes’ (i.e., 2–4 times within the past week), or ‘rarely (i.e., never or 1 time within the past week). A higher score reflects more significant sleep disturbances. Items are combined to form the following 8 subscales: bedtime resistance, sleep onset delay, sleep duration, sleep anxiety, night waking, parasomnias, sleep disordered breathing, and daytime sleepiness. A total score is calculated as the sum of all CSHQ scored items and can range from 33 to 99. A total score of 41 and above indicates a pediatric sleep disorder, as this cutoff has been shown to accurately identify 80% of children with a clinically diagnosed sleep disorder [34]. Parents were instructed to answer questions regarding their child’s sleep during a typical recent week. The questionnaire was completed at the onset and end of each treatment period. The completed CSHQ questionnaires were excluded from analysis if more than 20% of the data were missing. 

*Clinical Global Impression–Improvement scale (CGI-I)* [40] was used to measure the improvement in disruptive behaviors from the baseline. Scores range from 1 (very much improved), to 4 (unchanged), to 7 (very much worse). Scores of 1 or 2 (much improved) were defined as a positive response and all others indicated a negative response [40]. CGI-I was assessed at the end of each treatment period. Anchoring instructions were used to rate improvement in behavioral difficulties on the CGI-I, rather than improvement in overall ASD symptoms. The same clinician (AA) assessed and rated the CGI-S and CGI-I of all participants. Notably, while the CGI-S and CGI-I were developed to assess ‘overall function’, we used anchor points that were ‘domain-specific’ for disruptive behavior.

*Social Responsiveness Scale (SRS-2*): [41] this 65-item, caregiver questionnaire quantifies autism symptom severity (total scores range from 0 to 195, with higher scores indicating worsening severity). The questionnaire was completed at the onset and end of each treatment period.

### 2.7. Statistical Analyses

The impact of treatment on sleep was assessed using the change in CSHQ scores in each treatment period. Difference in the CSHQ total score was assessed both as a continuous and a dichotomous variable, using the cutoff score of 41. We adjusted for the following variables: sex, age at enrollment, and maternal education.

Continuous variables were assessed by two-tailed paired t-tests or ANOVA (after confirmation for normal/near normal distribution). Categorical variables were assessed by Pearson χ^2^ test. Treatment efficacy was compared between groups during the first and second treatment period and within treatment groups for participants who completed both treatment periods (per protocol [PP] analysis). Analyses were performed using IBM SPSS^®^ version 25 (2017). All *p* values were two-sided. *p*-value < 0.05 was considered significant.

## 3. Results

### 3.1. Participants 

Between 11 January 2017 and 12 April 2018, 150 children and adolescents (mean age 11.8 ± 4.1 years, median 11.25, range 5.1–20.8; 80% boys) entered the trial. The ASD symptoms were ‘severe’ in 78.7% per ADOS-2 (comparison score = 8–10) [31] and the adaptive levels were ‘low’ (composite score ≤ 70) in 88%, as per the Vineland Behavior Scales [32]. 

The participant’s characteristics are provided in Table 1. Fifty participants were randomly assigned to each of the three treatments in Period 1 and 44 participants per group completed the study (Figure 2). 

Among the 150 participants who underwent randomization, 131 (87%) submitted valid questionnaires at the onset and end of the first treatment period (Figure 2), enabling a between-subject analysis in this period (i.e., to compare the change in sleep parameters between the participants who received cannabinoids and the participants who received the placebo). In total, 107 participants (71%) submitted valid questionnaires at the onset and the end of both the first and second treatment period, enabling a within-subject analysis (i.e., to compare the change in sleep parameters while receiving cannabinoids, while receiving the placebo, and in participants who received both treatments).

The participants’ baseline characteristics, including sleep disturbances, as indicated by the CSHQ total and sub scores, were similar in the three treatment arms (Table 1).

Overall, 18 participants (12%) withdrew from the trial for the following reasons: 13 for reasons unrelated to treatment, three due to adverse events, and two due to ineffectiveness. In total, 131 participants (87%) had valid CSHQ scores before and after the treatment in the first treatment period. In total, 107 participants (71%) had valid pre-and post-treatment scores in both treatment periods, allowing a within-subject comparison.

### 3.2. Baseline Sleep Disturbances

Among the 146 participants who had valid CSHQ scores at the baseline, 125 (86%) had a CSHQ total score ≥41, indicating a sleep disorder. Higher CSHQ scores (indicating more prominent sleep disorder symptoms) at the baseline were correlated with a younger age (Pearson correlation r = −0.288, *p* < 0.001) and with higher SRS total scores, indicating more severe core autistic traits (r = 0.175, *p* = 0.036). The CSHQ scores were not associated with sex or adaptive behavior, as indicated by the VABS composite scores. 

Notably, the baseline characteristics were not different between the participants included in the per-protocol analysis and the participants who were excluded due to withdrawal or missing data, including age (*p* = 0.83); sex (*p* = 0.86); the severity of sleep disorders, as reflected by the CSHQ total score (*p* = 0.63); adaptive behavior, as evaluated by the VABS Composite scores (*p* = 0.57); and the severity of the core autistic symptoms, as assessed by the ADOS-2 (*p* = 0.58), CARS (*p* = 0.75), and the SRS (*p* = 0.25). 

### 3.3. Impact of Cannabinoid Treatment on Sleep

The impact of the cannabinoid treatment on sleep disturbances was assessed using the CSHQ. In total, 131 participants had valid CSHQ scores, both pre-treatment and post-treatment, in the first 12-week treatment period. Among these 131 participants, 44 received a whole-plant extract (BOL-DP-O-01-W, CBD:THC ratio = 20:1), 42 received pure cannabinoids (BOL-DP-O-01, CBD, and THC at a 20:1 ratio), and 45 received a placebo. The CSHQ total scores and the subscale scores did not differ significantly between the participants who received cannabinoids and the participants who received the placebo (Table 2). None of these measures differed significantly between the participants who received the whole-plant extract versus the pure cannabinoids (Table 2). 

Similar negative results were found in the second treatment period (Appendix A) and when comparing the two treatments that each participant received, using a within-participant analysis (Appendix A).

### 3.4. Longitudinal Associations between Sleep, Behavior, and Autistic Core Symptoms

Regardless of the treatment, improvements in the sleep disturbances, as indicated by a decline in the CSHQ total score, were associated with improvements in the autistic core symptoms, as well as the associated disruptive behaviors in both treatment periods. 

The autistic core symptoms were assessed using the SRS total score (higher scores indicate higher severity of symptoms). Changes in the SRS total score correlated with changes in the CSHQ total score in Period 1 (Pearson correlation: r = 0.266, *p* = 0.008) and Period 2 (r = 0.309, *p* = 0.004). 

Improvements in the ASD-associated disruptive behaviors were evaluated by the Clinical Global Impression–Improvement rate (CGI-I: lower rates indicate improvement). The CGI-I rate was associated with a change in the CSHQ total score in Period 1 (one-way ANOVA: f = 4.5, *p* = 0.013) and Period 2 (f = 3.36, *p* = 0.038).

## 4. Discussion

Interest in cannabis preparations as therapeutic agents in neuropsychiatric disorders is growing in both the scientific and lay communities [42,43]. This interest is particularly strong in disorders with substantial unmet needs, such as refractory epilepsy and pediatric ASD, which lacks medications that target its core symptoms [44]. Currently, robust evidence exists only for Epidiolex, a plant-derived pure CBD isolate, to treat the following specific types of refractory epilepsy: Dravet syndrome [45], Lennox–Gastaut syndrome [46], and tuberous sclerosis complex [47]. Nevertheless, full-spectrum extracts of various cannabis strains and synthetic cannabinoids are being widely used to treat adults living with chronic pain [48,49,50], chemotherapy-induced nausea and vomiting [51,52], sleep disorders [26], depression, anxiety, psychosis [53], PTSD [54], and to treat children with various types of refractory epilepsy [55], and irritability associated with autism spectrum disorder (ASD) [22,24,25,56]. 

The endocannabinoid system is involved in the pathophysiology of both sleep disorders [9] and ASD [19], which might contribute to the high incidence of sleep disturbances in people with ASD. These associations make the endocannabinoid system an attractive target for the treatment of sleep disturbances in ASD.

There is much anectodical evidence, as well as several reports of uncontrolled case series, suggesting an improvement in sleep disturbances following treatment with various strains of medical cannabis [24,25,57,58]. However, placebo-controlled studies have not been published so far.

In this randomized, placebo-controlled trial we used the following two CBD-rich preparations: a full-spectrum (whole-plant) extract, and purified CBD and THC isolates (pure cannabinoids). This is particularly relevant for sleep disturbances as it is commonly believed that, in addition to the main cannabinoids, other components of the cannabis plant, such as terpenes and flavonoids, also assist in alleviating sleep disturbances (an entourage effect) [59,60,61].

In our cohort, the effect of these two cannabinoid preparations at a dose of ~5.5 mg CBD and ~0.3 mg THC per kg, per day, was not superior to the placebo in all aspects of sleep measured by the CSHQ.

These findings are in line with a recent report on the negative acute effect of CBD on the sleep–wake cycle of healthy adults in a placebo-controlled study [62]. Of note, while a recent meta-analysis of randomized clinical trials demonstrated that cannabinoids provided a small benefit for impaired sleep, the studies that were analyzed were mainly in adults living with chronic pain, who used THC-rich preparations [26].

Accordingly, future studies of cannabinoid treatment for sleep disorders should consider using a more balanced CBD to THC ratio. Indeed, recreational cannabis strains that contain high THC and low CBD concentrations were associated with serious adverse events when used during youth, including decreased motivation [63,64,65], addiction [66], mild cognitive decline [64,67,68,69], and schizophrenia [64,70,71,72]. However, all of these risks are higher in cannabis strains with a high ratio of THC to CBD [73], than in the more balanced strains.

Consistent with previous studies [39,74], we also found that the severity of sleep disturbances, as indicated by the CSHQ total score at the baseline, correlated with a younger age and with the severity of the autistic core symptoms. Most of the participants in our cohort had severe autistic symptoms (78.7% had a comparison score of 8–10 in the ADOS-2). This might explain the higher rate of participants (86%) who screened positive for sleep disturbances, compared to a rate of approximately 70% that was reported in a US registry study [74] and in a Chinese multicenter survey [39]. 

Notably, we found a longitudinal association between changes in the CSHQ total score (in participants who received either cannabinoids or the placebo) and changes in the disruptive behavior and the severity of the core symptoms, suggesting the possibility of a cause-and-effect relationship (better sleep leads to lower symptoms). These findings are congruent with associations between sleep quality, behavior, and the severity of autistic core symptoms that have been reported in cross-sectional studies [39,75]. This finding also underscores the importance of sleep quality in children with ASD, the impact of which may exceed that of their typically developing peers.

Our study had several limitations: The study was designed as a cross-over study, which allows within-participant analyses, comparing the two treatments that each participant received. However, a treatment order effect (all treatments were more effective in the first period, probably due to a greater initial placebo effect) made this analysis less accurate. Therefore, we reported the more accurate between-subject analyses of the first treatment period (Table 2). We also present, in the Appendix A, the within-participant analyses (Appendix A) and the between-participant analyses of Period 2 (Appendix A), which yielded similar negative results. Another limitation of this study was the use of a caregiver’s report for assessing sleep quality, without more objective measures of sleep such as actigraphy and sleep logs. Additionally, our study was not powered to detect the effects of age, the level of function, and other baseline characteristics on the treatment response.

## 5. Conclusions

Sleep disturbances are very common in children with ASD, and they have a substantial impact on the quality of life of the child and the family. Preliminary clinical evidence and preclinical studies, which implicate the endocannabinoid system in the pathophysiology of both ASD and sleep disorders, suggest that cannabinoid treatment might improve sleep in children with ASD. In a controlled study of 150 participants, we found that a whole-plant extract and a pure cannabinoid preparation, which contained CBD and THC in a 20:1 ratio did not improve the sleep parameters, as reflected in the CSHQ scores. Future studies should consider using actigraphy and sleep logs and recruiting participants within narrower ranges of age and functional levels, this might enable the identification of target populations within the autism spectrum that might benefit from this line of treatment.

## Figures and Tables

**Figure 1 biomedicines-10-01685-f001:**
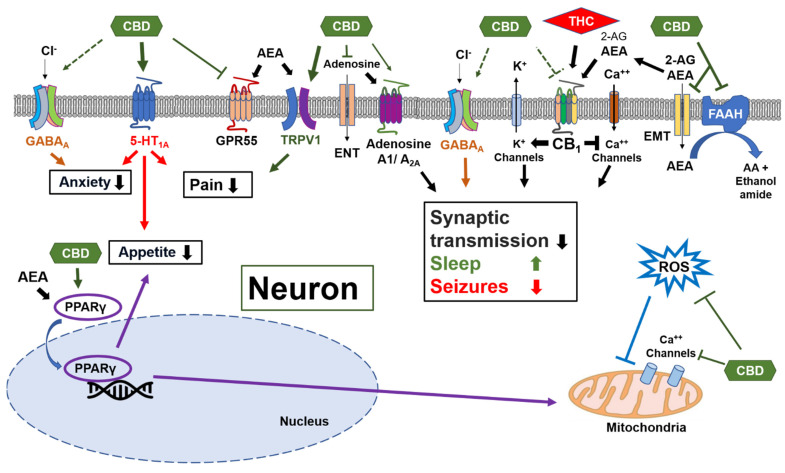
Multiple molecular targets for cannabidiol (CBD) in neurons.

**Figure 2 biomedicines-10-01685-f002:**
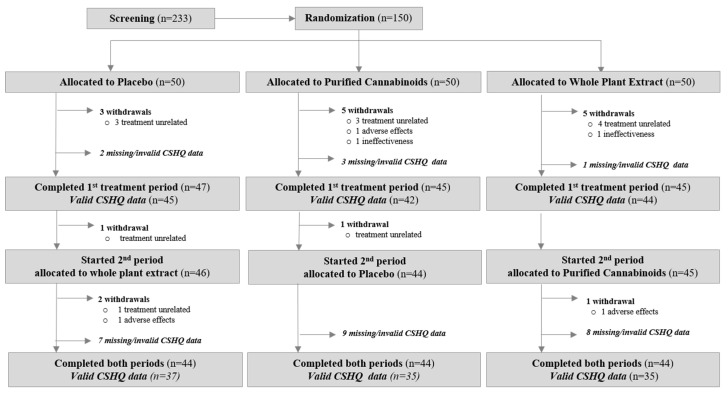
Participants’ allocation and adherence.

**Table 1 biomedicines-10-01685-t001:** Participants’ baseline characteristics.

	All (n = 150)	Group A (n = 50)	Group B (n = 50)	Group C (n = 50)	Sig.
**Treatment 1st period**		Placebo	Pure cannabinoids	Whole plant	
**Treatment 2nd period**		Whole plant	Placebo	Pure cannabinoids	
**Sex**					
Males **n** (%)	**120** (80%)	**42** (84%)	**42** (84%)	**36** (72%)	**0.22** *
**Age**					
**Mean** ± SD [median, range]	**11.8** ± 4.1 [11.3, 5.1–20.8]	**11.7** ± 3.8 [10.7, 5.8–20.0]	**11.6** ± 4.3 [10.3, 5.1–20.4]	**12.1** ± 4.3 [12.6, 5.1–20.8]	**0.79** ^#^
**BMI**					
**Mean** ± SD [median, range]	**20.8** ± 5.7 [19.0, 12.3–39.6]	**20.5** ± 5.2 [19.1, 12.8–34.0]	**20.5** ± 6.0 [19.1, 12.3–39.6]	**21.3** ± 6.1 [19.0, 13.9–39.5]	**0.72** ^#^
**ADOS comparison score**					
**Mean** ± SD [median, range]	**8.8** ± 1.5 [10.0, 4.0–10.0]	**8.6** ± 1.6 [9.0, 4.0–10.0]	**9.2** ± 1.3 [10.0, 6.0–10.0]	**8.6** ± 1.6 [9.0, 4.0–10.0]	**0.07** ^#^
**VABS composite score**					
**Mean** ± SD [median, range]	**52.3** ± 14.5 [51.0, 20.0–102.0]	**52.0** ± 15.0 [49.0, 26.0–102.0]	**52.4** ± 15.2 [54.0, 25.0–89.0]	**52.3** ± 13.6 [52.0, 20.0–78.0]	**0.99** ^#^
**CARS total score**					
**Mean** ± SD [median, range]	**45.4** ± 8.4 [47.5, 29.5–59.0]	**46.0** ± 8.5 [47.3, 30.5–59.0]	**45.5** ± 8.9 [48.5, 29.5–57.5]	**44.6** ± 7.8 [46.5, 31.0–56.5]	**0.68** ^#^
**SRS**					
**Mean** ± SD [median, range]	**119** ± 27 [121, 53–180]	**122** ± 23 [124, 53–159]	**118** ± 31 [118, 64–178]	**117** ± 27 [117, 66–180]	**0.37** ^#^
**Concomitant medications**					
Atypical antipsychotics n (%)	**76** (50.7%)	**28** (56.0%)	**20** (40.0%)	**28** (56.0%)	**0.18** *
Typical antipsychotics n (%)	**13** (8.7%)	**5** (10.0%)	**3** (6.0%)	**5** (10.0%)	**0.82** *
Anticonvulsants n (%)	**18** (8.7%)	**6** (12.0%)	**4** (8.0%)	**8** (16.0%)	**0.47** *
Stimulants n (%)	**20** (13.3%)	**5** (8.0%)	**11** (22.0%)	**5** (10.0%)	**0.08** *
Benzodiazepines n (%)	**5** (3.3%)	**1** (2.0%)	**2** (4.0%)	**2** (4.0%)	**1.00** *
Melatonin n (%)	**12** (8.0%)	**6** (12.0%)	**4** (8.0%)	**2** (4.0%)	**0.39** *
SSRIs n (%)	**21** (14.0%)	**6** (12.0%)	**8** (16.0%)	**7** (4.0%)	**0.84** *
**Total CSHQ score**					
**Mean** ± SD [median, range]	**49.9** ± 9.2 [48.5, 34.0–73.5]	**49.7** ± 8.7 [49.0, 34.0–69.2]	**50.1** ± 9.4 [47.5, 36.0–72.0]	**49.7** ± 9.6 [35.0, 34.0–73.5]	**0.97** ^#^
**Bedtime Resistance**					
**Mean** ± SD [median, range]	**9.4** ± 3.3 [8.0, 6.0–17.0]	**9.7** ± 3.2 [9.0, 6.0–17.0]	**9.4** ± 3.4 [6.0–16.0]	**9.1** ± 3.3 [8.0, 6.0–17.0]	**0.63** ^#^
**Sleep Onset Delay**					
**Mean** ± SD [median, range]	**1.9** ± 0.8 [2.0, 1.0–3.0]	**1.9** ± 0.8 [2.0, 1.0–3.0]	**1.9** ± 0.9 [2.0, 1.0–3.0]	**2.0** ± 0.8 [2.0, 1.0–3.0]	**0.99** ^#^
**Sleep Duration**					
**Mean** ± SD [median, range]	**4.6** ± 1.8 [4.0, 3.0–9.0]	**4.6** ± 1.8 [4.0, 3.0–9.0]	**4.6** ± 1.8 [4.0, 3.0–9.0]	**4.5** ± 1.8 [4.0, 3.0–9.0]	**0.96** ^#^
**Sleep Anxiety**					
**Mean** ± SD [median, range]	**6.3** ± 2.3 [6.0, 4.0–12.0]	**6.4** ± 2.1 [6.0, 4.0–11.0]	**6.5** ± 2.4 [6.0, 4.0–12.0]	**6.1** ± 2.3 [5.0, 4.0–12.0]	**0.64** ^#^
**Night Wakings**					
**Mean** ± SD [median, range]	**4.9** ± 1.9 [4.0, 3.0–9.0]	**4.8** ± 1.7 [4.5, 3.0–9.0]	**5.13** ± 2.0 [5.0, 3.0–9.0]	**4.7** ± 2.0 [4.0, 3.0–9.0]	**0.52** ^#^
**Parasomnias**					
**Mean** ± SD [median, range]	**9.2** ± 2.0 [9.0, 7.0–18.2]	**8.6** ± 1.7 [9.0, 7.0–12.0]	**9.5** ± 2.0 [9.0, 7.0–14.0]	**9.3** ± 2.3 [8.8, 7.0–18.2]	**0.44** ^#^
**Sleep Disordered Breathing**					
**Mean** ± SD [median, range]	**3.9** ± 1.4 [3.0, 3.0–9.0]	**3.9** ± 1.2 [3.5, 3.0–7.0]	**3.7** ± 1.3 [3.0, 3.0–9.0]	**4.1** ± 1.6 [3.0, 3.0–7.0]	**0.40** ^#^
**Daytime Sleepiness**					
**Mean** ± SD [median, range]	**14.4** ± 3.6 [14.0, 9.0–24.0]	**14.2** ± 3.7 [14.0, 9.0–23.0]	**14.3** ± 3.7 [13.1, 9.0–24.0]	**14.7** ± 3.5 [15.0, 9.0–24.0]	**0.79** ^#^

Baseline characteristics of participants stratified to treatment arms. ADOS-2—Autism Diagnostic Observation Schedule, comparison score of 8–10 indicated severe autistic symptoms; BMI—body mass index; CARS—Childhood Autism Rating Scale, scores above 36.5 are indicative of severe ASD; CSHQ—Children’s Sleep Habits Questionnaire; SRS—Social Responsiveness Scale, total score ≥ 75 indicates severe autistic symptoms; VABS—Vineland Adaptive Behavior Scale, composite score ≤ 70 indicates low adaptive level. * Categorical parameters (sex and medications) were compared using Pearson chi-square tests. ^#^ Continuous parameters were compared using one-way analysis of variance (ANOVA).

**Table 2 biomedicines-10-01685-t002:** Impact of cannabinoid treatment on sleep. Comparison of treatment effects in the 1st 12-week period.

	Placebo n = 45 [Change in Points]	Pure Cannabinoids n = 42 [Change in Points]	Whole Plant n = 44 [Change in Points]	Total n = 131 [Change in Points]	Sig ^^^
**Total CSHQ score**					
**Mean** ± SD [median, range]	**−1.4** ± 6.6 [−1.9, −20.3–13.0]	**−2.9** ± 9.2 [−1.5, −27.9–18.0]	**−2.3** ± 5.6 [−1.5, −18.0–7.3]	**−2.2** ± 7.2 [−1.9, −27.9–18.0]	**0.63**
**Bedtime Resistance**					
**Mean** ± SD [median, range]	**−0.6** ± 1.6 [0.0, −4.0–3.0]	**−0.5** ± 2.7 [0.0, −9.0–5.7]	**−0.3** ± 1.6 [0.0, −6.0–3.0]	**−0.4** ± 2.0 [0.0, −9.0–5.7]	**0.79**
**Sleep Onset Delay**					
**Mean** ± SD [median, range]	**−0.1** ± 0.6 [0.0, −1.0–2.0]	**−0.1** ± 0.8 [0.0, −2.0–2.0]	**−0.2** ± 0.8 [0.0, −2.0–1.0]	**−0.2** ± 0.7 [0.0, −2.0–2.0]	**0.98**
**Sleep Duration**					
**Mean** ± SD [median, range]	**−0.1** ± 1.6 [0.0, −4.0–4.0]	**0.0** ± 2.0 [0.0, −5.0–4.0]	**−0.5** ± 1.9 [0.0, −5.0–4.0]	**−0.2** ± 1.8 [0.0, −5.0–4.0]	**0.38**
**Sleep Anxiety**					
**Mean** ± SD [median, range]	**−0.4** ± 1.2 [0.0, −4.0–2.0]	**−0.6** ± 1.3 [0.0, −4.0–1.7]	**−0.2** ± 1.5 [0.0, −4.0–2.0]	**−0.4** ± 1.3 [0.0, −4.0–2.0]	**0.59**
**Night Wakings**					
**Mean** ± SD [median, range]	**−0.2** ± 1.3 [0.0, −3.0–3.0]	**−0.8** ± 1.5 [−0.5, −4.0–1.0]	**−0.6** ± 1.2 [0.0, −4.0–1.0]	**−0.5** ± 1.4 [0.0, −4.0–3.0]	**0.11**
**Parasomnias**					
**Mean** ± SD [median, range]	**−0.2** ± 1.6 [0.0, −4.0–4.0]	**−0.6** ± 1.9 [−0.9, −7.0–4.0]	**−0.5** ± 1.4 [0.0, −4.5–2.3]	**−0.5** ± 1.6 [0.0, −7.0–4.0]	**0.53**
**Sleep Disordered Breathing**					
**Mean** ± SD [median, range]	**−0.0** ± 0.9 [0.0, −2.0–3.0]	**−0.3** ± 1.0 [−0.0, −4.0–1.0]	**−0.1** ± 0.8 [0.0, −2.0–1.0]	**−0.2** ± 0.9 [0.0, −4.0–3.0]	**0.36**
**Daytime Sleepiness**					
**Mean** ± SD [median, range]	**0.1** ± 3.0 [0.0, −9.0–7.8]	**0.2** ± 3.5 [0.0, −7.0–7.0]	**0.0** ± 2.7 [0.0, −5.0–5.0]	**0.1** ± 3.1 [0.0, −9.0–7.8]	**0.96**

Between-subject analyses of the change in the CSHQ scores following treatment in the first treatment period. CSHQ—Children’s Sleep Habits Questionnaire. Positive change (increment of CSHQ scores) indicates worsening of the sleep disorder. Change in the CSHQ scores from baseline following treatment is compared between the 3 treatment arms. ^ One-way ANOVA for influence of treatments between study groups. Notably, the difference between cannabinoid treatment and placebo was not statistically significant, even when combining the two cannabinoid treatments into one group, compared to placebo (data not shown).

## Data Availability

The authors declare that the data supporting the findings of this study are available within the paper and its Appendix A. The remainder of the data are available from the corresponding author upon reasonable request.

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
