# Peer review of "A Placebo-Controlled Trial of Cannabinoid Treatment for Disruptive Behavior in Children and Adolescents with Autism Spectrum Disorder: Effects on Sleep Parameters as Measured by the CSHQ"

_biomedicines, 2022, doi:10.3390/biomedicines10071685_

Round 1

Reviewer 1 Report

This is a report from a clinical trial that investigated the safety and efficacy of using cannabis to treat sleep disturbance in children, adolescents, and young adults with ASD.  While it is true there is a wide age range of participants, the sample size of 150 should adequate to account for age related differences in presentation and response to treatment. The clinical trial was a double-blind, placebo controlled trial, which is the gold standard of science. The I/E were appropriate for this population and adhered to an "industry standard."  Measures were appropriate for this population.  Results, while surprising and a little disheartening, were produced from the appropriate statistical analyses and the discussion is well supported by the results.  This study represents an important contribution to the literature as cannabis is widely used by this population as a treatment option as it becomes more readily available in the US.  I suspect we will many derivatives of cannabis being tested in the scientific literature in the years to come. 

Abstract: Line 26.  Questionnaire is misspelled as "Quetioner." Interesting association between CSHQ and SRS scores - most likely attributed to placebo effect.  This finding is also important for the literature. 

Introduction: P2 line 65. The sentence that begins with "Alterations" should be the last sentence of the previous paragraph and not its own paragraph. Figure 1 is excellent. 

Materials and Methods:  The use of the CT identifier negatively affects readability.  It would be helpful to add one sentence to explain the perceived difference between the two cannabis formulations.  That is, is one more potent then the other? Absorbed more easily?  

Good that you adjusted for sex and age in the analysis. All analyses are appropriate. 

Results:  Why were there only 131 valid questionnaires?  Might be better to present the withdrawal paragraph (p7, line 217) before you discuss the valid data to be analyzed.  Also, 150 - 18 = 132.   You are missing 1 questionnaire.

Discussion: Seems to present more background literature rather than a discussion of the results from this study.  In the investigators opinion, was the concentration of TCH too low to have an effect? The discussion highlights SAEs associated with cannabis with citations, but they seem inflated without the actual statistics in the cited articles.  For example, what % experienced cognitive decline and over what period of time?  And schizophrenia? Are these real concerns for using cannabis?  What were the concentrations of THC in these studies and can we use this information to find a sweet spot for effective treatment?  That is, is there a higher concentration than 20:1 that is perhaps less than the concentration associated with the SAEs that might be used in the next investigation?  This potential should be discussed even if it is speculative at this point. 

Also, the improvement in core symptoms (SRS scores).  Is that real or placebo?  Seems more likely associated with placebo, but this should be fleshed out a bit more.  If the investigators dont think it is placebo effect, then why?

All in all, an excellent study and manuscript. However, more information is needed in the discussion to help the science in this field move forward and build on what you've done.  Even if speculative, the field can benefit from your viewpoint and expertise. 

Reviewer 2 Report

This paper by Schnapp et al. provides data from a double-blind placebo-controlled study aimed at evaluating possible beneficial effects of CBD: THC combinations (in a ratio 20:1 either as full-spectrum extract or as pure cannabinoids) on sleep disturbances in ASD patients aged 5 to 21 years. Findings from this clinical study show that cannabinoid combinations failed to improve sleep disturbances in ASD patients when assessed through the CSHQ. However, data seem to suggest a slight improvement in the SRS total score, which prompt further clinical evaluation of the efficacy of CBD-rich preparations on core ASD symptoms.

Methodology is sound and participants’ baseline characteristics are similar among treatment arms. I appreciated the inclusion of a comparison between the effects of whole-cannabis extracts and pure cannabinoids in the study and the discussion of study limitations provided by the Authors at the end of the manuscript.

Minor comments:

1)    In the abstract, the Authors clearly state that “CBD-rich cannabinoid treatment was not superior to placebo in all aspects of sleep measured by the CSHQ, ... Notably, regardless of treatment, improvement in the CSHQ total-score was associated with improvements in the autistic core-symptoms…”. This is also reported in the Introduction (page 2, lines 79-81). The two statements in their present form sound contradictory. Please, explain better in order to clarify the Authors’ intended meaning and to avoid any confusion.

2)    Please check carefully for typos throughout the whole manuscript:

-       Punctuation should be placed after the closing parenthesis (whole manuscript)

-       Extra spaces (page 1 line 41, page 2 line 58, page 3 line 97, …)

-       Page 3 line 95: “5-hidroxytriptamine” should be “5-hydroxytriptamine”
